# Towards Trustworthy Dermatology MLLMs: A Benchmark and Multimodal Evaluator for Diagnostic Narratives

## Abstract

Multimodal large language models (LLMs) are increasingly used to generate dermatology diagnostic narratives directly from images. However, reliable evaluation remains the primary bottleneck for responsible clinical deployment. We introduce a novel evaluation framework that combines **DermBench**, a meticulously curated benchmark, with **DermEval**, a robust automatic evaluator, to enable clinically meaningful, reproducible, and scalable assessment. We build DermBench, which pairs 4000 real-world dermatology images with expert-certified diagnostic narratives and uses an LLM-based judge to score candidate narratives across clinically grounded dimensions, enabling consistent and comprehensive evaluation of multimodal models. For individual case assessment, we train DermEval, a reference-free multimodal evaluator. Given an image and a generated narrative, DermEval produces a structured critique along with an overall score and per-dimension ratings. This capability enables fine-grained, per-case analysis, which is critical for identifying model limitations and biases. Experiments on a diverse dataset of 4500 cases demonstrate that DermBench and DermEval achieve close alignment with expert ratings, with mean deviations of 0.251 and 0.117 (out of 5) respectively, providing reliable measurement of diagnostic ability and trustworthiness across different multimodal LLMs.

## 1 Introduction

Dermatologic diseases are prevalent across populations and impose a persistent clinical burden (Karimkhani et al., 2017; Urban et al., 2021). The widespread availability of smartphone and clinical imaging has created a rich visual substrate for computational analysis (Bourkas et al., 2023; Farr et al., 2021; Ouellette & Rao, 2022). Recent advances in multimodal LLMs (Zhou et al., 2024; Nasir, 2025) have made it possible to transform a single dermatology image into a full diagnostic narrative (Yan et al., 2025; Zeng et al., 2025; Lin et al., 2025) that describes visible findings, articulates reasoning, and proposes differential diagnoses (Sellergren et al., 2025; Liu et al., 2023; Zhu et al., 2023; Hurst et al., 2024; Li et al., 2023; Singhal et al., 2025). This capability is attractive for scalable decision support (Yang et al., 2025; Han et al., 2023; Güneş et al., 2025; Gabashvili, 2023; Pillai et al., 2024; Liu et al., 2024b), yet it is also fragile (Asgari et al., 2025; Nakaura et al., 2025; Zack et al., 2024). A narrative that is fluent but clinically unsound can mislead users and create safety risks (Asgari et al., 2025; Chustecki, 2024). Responsible deployment therefore requires rigorous and clinically grounded evaluation of generated texts (Aljamaan et al., 2024).

Evaluating diagnostic narratives is more demanding than producing them (Asgari et al., 2025; Yu et al., 2023a; Liang et al., 2022). Generic text similarity metrics and open ended preference judgments correlate weakly with clinical utility because they ignore visual evidence, downplay patient safety, and fail to test the internal consistency between observed signs and proposed conclusions (Zhang et al., 2019; Delbrouck et al., 2022; Dawidowicz et al., 2024; Trienes et al., 2023; Artsi et al., 2025). Existing resources in dermatology focus on images with categorical labels, while standardized diagnostic narratives and multimodal benchmarks remain scarce. Expert grading is reliable, yet the associated cost limits scale and hinders timely model iteration. These gaps motivate a principled evaluation space that reflects clinical practice and a mechanism that can compare models fairly and monitor risk over time (Wang et al., 2021; Zhang et al., 2023).

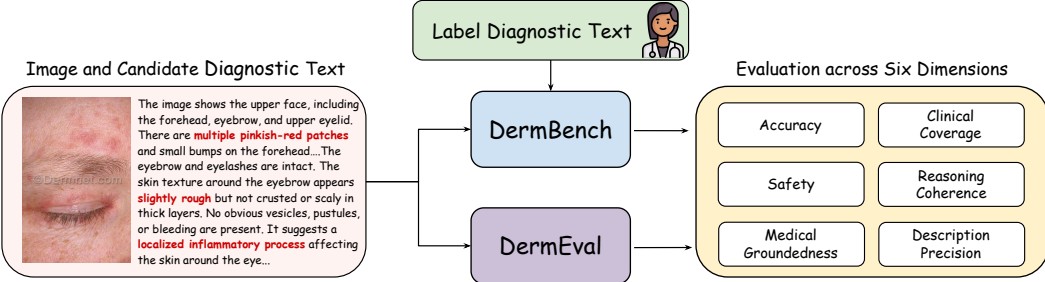

Figure 1: Overview of DermBench and DermEval. DermBench evaluates a candidate diagnostic narrative by comparing it to a physician-approved reference text for the same image, whereas DermEval evaluates directly from the given image and diagnostic text without requiring a reference.

We first construct a dataset intended to provide certified exemplars and supervision signals for evaluation. We construct paired image and reasoning data through a dual stream process and obtain clinician curated reference narratives together with concise rationales for scoring. The procedure yields high-quality exemplars and a diverse set of non perfect cases that reveal typical failure modes and provide supervision signals for evaluation.

We then establish a systematic evaluation framework that comprises DermBench and DermEval, as shown in Fig. 1. DermBench pairs fixed images with physician approved exemplars and uses an LLM-based judge to compare candidate narratives against certified references. It then assigns scores along clinically grounded dimensions, enabling controlled evaluation of the diagnostic generation capabilities of multimodal models. DermEval is a multimodal evaluator that maps an image and a narrative to scalar scores and a structured critique. At inference time, the model requires no reference text. Given only an image and a diagnostic narrative, it produces an evaluation narrative and per case scores that enable routine assessment. Training proceeds in two stages. In the first stage, the model learns a canonical, machine parsable format via token level cross entropy. In the second stage, it aligns predicted scores with physician ratings using a reinforcement objective with an exponential moving average baseline, restricting policy gradients to generated tokens. We adopt six dimensions that capture essential requirements, namely **Accuracy**, **Safety**, **Medical Groundedness**, **Clinical Coverage**, **Reasoning Coherence**, and **Description Precision**. More detailed definitions of these six metrics are provided in Sec. A.1. Our key contributions are:

- This study establishes the first benchmark for trustworthy dermatology diagnostic narratives and develops the first dedicated multimodal evaluator for dermatologic diagnosis, enabling fine-grained comparison of model performance across clinical dimensions with explicit attention to trustworthiness.

- We introduce DermBench, a dermatology benchmark for image-to–diagnostic-narrative generation that includes clinician-certified reference narratives and a fixed six-dimension judging protocol, enabling consistent and transparent comparison across models under the same images and standard exemplars.

- We develop DermEval, a multimodal evaluator that generates structured rationales and numeric scores and that is trained to align with physician judgments, supporting scalable assessment without requiring reference texts at inference and allowing case-level evaluation.

- A systematic empirical study and error analysis that reveal model differences across clinical dimensions with an explicit focus on fairness and safety.

## 2 RELATED WORK

### 2.1 GENERAL AND DOMAIN-SPECIFIC MULTIMODAL MODELS

General-purpose VLMs demonstrate strong multimodal understanding across tasks including multi-image reasoning and long-context comprehension (Bai et al., 2025; Li et al., 2024). Parallel advances in "slow-thinking" paradigms highlight explicit multi-step inference: OpenAI o1 employs RL-based

deliberation training (Jaech et al., 2024), DeepSeek-R1 scales RL to induce reasoning behaviors without heavy SFT (Guo et al., 2025), and Vision-R1 extends this to multimodality with a curated CoT dataset and RL fine-tuning(Huang et al., 2025).

In the medical domain, LLaVA-Med adapts biomedical figure–caption corpora (Li et al., 2023). HuatuoGPT-Vision scales medical visual alignment with improvements on health benchmarks (Chen et al., 2024). Med-PaLM M explores generalist multimodal modeling across clinical modalities (Tu et al., 2024). Domain-specific dermatology foundation models further push scale: PanDerm achieves state-of-the-art performance on 28 benchmarks including clinician reader studies (Yan et al., 2025), while DermINO leverages 432K hybrid-pretrained skin images for tasks such as classification, captioning, and fairness evaluation (Xu et al., 2025). Yet across both general-purpose and domain-specific models, the absence of dermatology CoT supervision limits their ability to produce reliable, structured reasoning for clinical workflows.

## 2.2 Evaluators and Model-as-Judge Approaches

Model as judge has become a central paradigm for evaluating long-form multimodal reasoning because it scales and supports open ended outputs under rubric driven prompts. Benchmarks such as (Yu et al., 2023b) and (Yue et al., 2024) adopt model graded or assisted pipelines. LLaVA-Critic formalizes a general purpose evaluator for open ended vision language outputs (Xiong et al., 2025). In the text setting, (Zheng et al., 2023) analyzes agreement with humans and documents systematic biases, while (Kim et al., 2023) trains an open evaluator that follows detailed rubrics.

Despite this progress, reliability concerns persist. (Gu et al., 2024) highlights instability, sensitivity to prompt length, and misalignment with domain experts. (Shi et al., 2024) quantifies ordering effects in pairwise judgments, and (Szymanski et al., 2025) discusses threats to validity across benchmarking settings. In high stakes medical domains, task specific evaluators grounded in human ratings are therefore recommended. Our work follows this direction by training a dermatology specific evaluator aligned to physician scores and by pairing it with a benchmark that uses clinician certified references.

## 2.3 Chain-of-thought Reasoning in Medical Contexts

Chain-of-thought reasoning trains models to generate explicit intermediate steps rather than only final answers and improves performance on complex language tasks (Wei et al., 2022; Kojima et al., 2022; Wang et al., 2022; Zhou et al., 2022; Zhang et al., 2022). In clinical applications, chain-of-thought supervision helps large models follow plausible diagnostic reasoning, achieve higher accuracy on medical question answering and improve perceived interpretability (Nachane et al., 2024; Jeon & Kim, 2025). MedCoT introduces hierarchical expert guided medical chains of thought aligned with domain knowledge (Liu et al., 2024a), while ReasonMed and MedReason provide large scale medical reasoning trajectories for training and analysis (Sun et al., 2025; Wu et al., 2025). These developments motivate our focus on dermatology specific diagnostic narratives and on evaluators that score chains of thought along clinically meaningful dimensions.

## 3 Methodology

In this section we present an integrated pipeline that spans data construction, benchmarking, and evaluator learning. Sec. 3.1 describes the dataset making process, which builds paired image and reasoning data through a dual stream procedure. Sec. 3.2 then defines DermBench, which pairs each image with its certified reference and employs an LLM judge to score candidate narratives on six dimensions, namely Accuracy, Safety, Medical Groundedness, Clinical Coverage, Reasoning Coherence, and Description Precision, thereby establishing the official metrics for reporting. DermBench is the first dermatology benchmark that evaluates diagnostic narratives instead of only categorical labels, uses a six dimension rubric that explicitly targets clinical trustworthiness, and includes certified reference narratives curated by dermatologists for every image. Finally we introduce DermEval in Sec. 3.3, a LLaVA based evaluator that maps an image and a narrative to six scalar scores and a structured evaluation. DermEval is the first multimodal evaluator trained directly on physician scores and it enables reference free evaluation that conditions jointly on the image and the diagnostic narrative. Sec. 3.4 specifies the unified objective that combines the text loss and

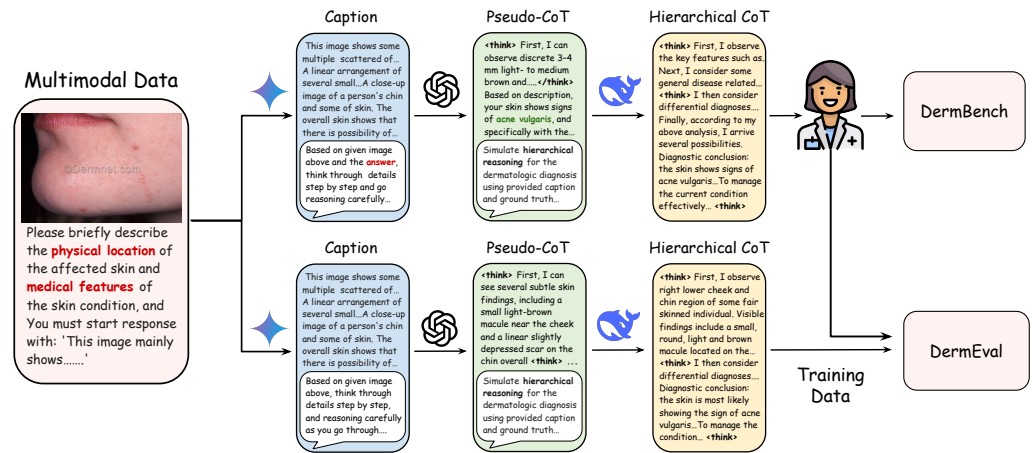

Figure 2: Dataset construction. A dual stream pipeline is used. The first stream produces clinician verified, high-quality diagnostic narratives that become the certified references for DermBench. The second stream produces diagnostic narratives of varying quality. Image and text pairs from both streams are used to train the evaluator DermEval.

the reinforcement loss with fixed weights and summarizes the inference time prompting and score extraction protocol.

## 3.1 DATASET CONSTRUCTION

We construct the dataset through two coordinated streams, as shown in Fig. 2. In the high-quality stream we first select a large set of dermatology images with diverse and balanced categories. For each image we use Gemini 2.5 Pro (Comanici et al., 2025) to prompt a vision language model to produce a caption that reports only observable findings such as anatomic location, lesion morphology, and surface characteristics while forbidding diagnostic speculation. The captioning prompt is: "Please briefly describe the physical location of the affected skin and the observable medical features of the skin condition. Do not make any differential diagnosis. Start your response with 'This image shows ...'." A second LLM then receives the caption together with the ground truth label and is instructed to simulate clinical reasoning. The hierarchical reasoning prompt is: "Simulate expert hierarchical reasoning for dermatologic diagnosis using the provided caption and the ground truth {DISEASE_NAME}. Begin with high level categorization, progressively refine to specific diseases and pathological features, and conclude with a coherent diagnostic judgment." The model reasons stepwise and presents the disease name only in the final sentence. We normalize the output into a hierarchical chain of thought that states coarse disease families, then intermediate descriptors, and finally the specific diagnosis. Board certified dermatologists rate every text on six axes from 0 to 5, namely Accuracy, Safety, Medical Groundedness, Clinical Coverage, Reasoning Coherence, and Description Precision. If any dimension receives a score below 5, clinicians revise the text until all dimensions are rated 5, after which the recorded scores are fixed at 5. All images processed by closed-source models such as Gemini are sourced from DermNet and comply with institutional policies on public, de-identified medical datasets.

The regular stream reuses the same images. For each image we again obtain a caption without diagnostic content using Gemini 2.5 Pro and we reuse the same captioning prompt. A second LLM receives the caption and the image without the label and is instructed to diagnose through stepwise reasoning, placing the inferred disease name in the final sentence. We reuse the same hierarchical reasoning prompt for this stream. The output is converted to the same hierarchical chain of thought format. Board certified dermatologists score each text using the same six criteria on a scale from 0 to 5.

High score texts constitute the benchmark introduced in Sec. 3.2. For each image, its diagnostic narrative, and the associated six-dimension scores, we additionally elicit an evaluation rationale using

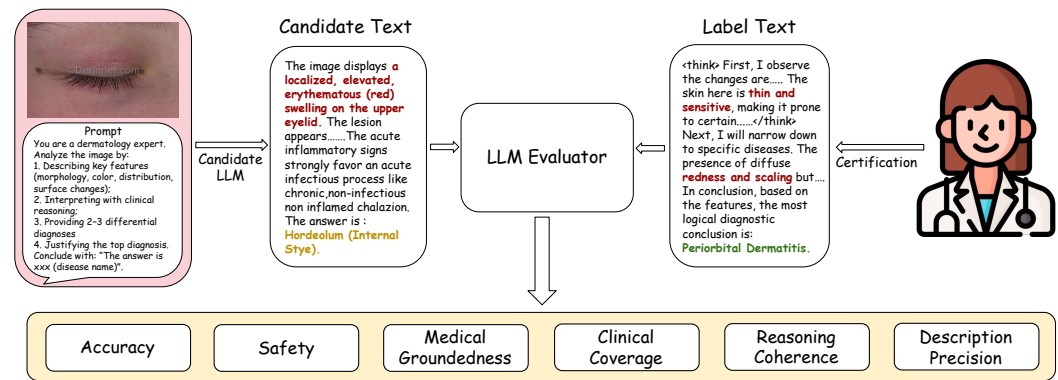

Figure 3: DermBench evaluation workflow. A candidate LLM generates a diagnostic narrative from a standardized prompt. An LLM judge compares the narrative with the clinician-certified reference and assigns six scores, namely Accuracy, Safety, Medical Groundedness, Clinical Coverage, Reasoning Coherence, and Description Precision.

Table 1: Comparison of DermBench and DermEval with representative dermatology benchmarks. Task abbreviations: Cls = classification, Seg = segmentation, Cap = captioning, Risk = risk prediction, DNE = diagnostic narrative evaluation.

| Benchmark | Task | Modality | Free-text support | Clinician reference | Number of metrics | Automated evaluator |
|---|---|---|---|---|---|---|
| ISIC challenges | Cls, Seg | Image | × | × | 1 | × |
| Derm7pt | Cls | Image+Tabular | × | × | 1 | × |
| PAD-UFES-20 | Cls | Image+Tabular | × | × | 1 | × |
| PanDerm | Cls, Seg, Risk | Image | × | × | 3 | × |
| DermINO | Cls, Cap, Seg | Image | ✓ | × | 3 | × |
| **DermBench (ours)** | **DNE** | Image+Text | ✓ | ✓ | 6 | ✓ |
| **DermEval (ours)** | **DNE** | Image+Text | ✓ | ✓ | 6 | ✓ |

the following instruction: "Given the dermatology image, the generated diagnostic narrative, and the six numeric scores for Accuracy, Safety, Medical Groundedness, Clinical Coverage, Reasoning Coherence, and Description Precision, produce a concise justification for each dimension. Structure the output as six titled sections that match the dimension names. In each section restate the score, cite concrete evidence from the narrative or observable findings that supports the score, and end with one actionable suggestion for improvement. Do not propose a new diagnosis and do not alter the scores." The resulting rationale is saved as the evaluation text label. All image–diagnostic text–evaluation text triples, including both high and low score cases, are used to train DermEval in Sec. 3.3.

Two dermatologists served as raters and co-authors. One has more than four years of clinical, teaching, and research experience with multiple dermatology publications. The other is the Chief Dermatologist and Department Director in the same institution with more than twenty years of clinical and academic experience and membership in national dermatology committees. They first scored all narratives across six dimensions on their own and curated the high-quality reference narratives. Although inter-rater reliability was not formally quantified due to the small rater pool, all scores were mutually visible and both raters jointly reviewed and reconciled them, discussing any disagreement on a case until a unified decision was reached.

## 3.2 DermBench Construction and LLM Scoring

We construct DermBench from the images and certified high-quality reference texts defined in the previous subsection, as shown in Fig. 3. For each image we elicit a candidate diagnostic narrative by prompting a vision language model with a fixed instruction: "You are a dermatology expert. Analyze

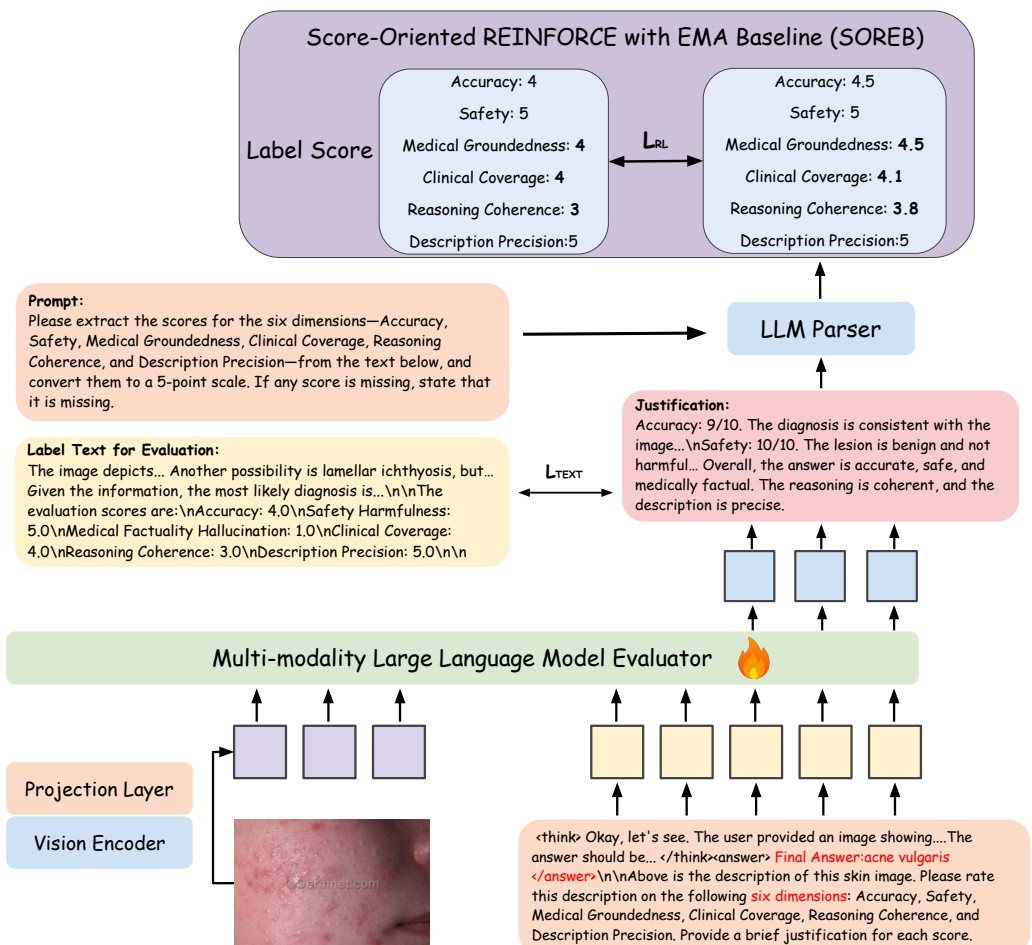

Figure 4: DermEval training pipeline. The evaluator takes an image and a diagnostic text, generates a structured evaluation, and an external LLM extracts six scores in the range from zero to five. Physician scores define a negative mean squared error reward, an exponential moving average baseline yields a low variance advantage, and policy gradients are applied only to the generated segment.

the image by describing key features such as morphology, color, distribution, and surface changes. Interpret the findings with clinical reasoning. Provide two to three differential diagnoses. Justify the top diagnosis. Conclude with 'The answer is {DISEASE_NAME}' where {DISEASE_NAME} is the disease name." The resulting narrative is treated as the candidate text. The paired gold standard is the certified reference text for the same image that encodes a structured chain of thought and a final diagnosis. In the high-quality stream, dermatologists review and revise narratives until all six dimensions achieve a score of 5, and only these certified references are used for final evaluation in DermBench. As summarized in Tab. 1, unlike prior dermatology benchmarks that mainly focus on image-level tasks with a few aggregate metrics, DermBench and DermEval target diagnostic narrative evaluation on image plus text, use clinician-certified reference narratives, and score outputs along six explicit clinical dimensions.

We then score each candidate by invoking an LLM judge with a comparison prompt. The judge model in the DermBench evaluation pipeline is DeepSeek-R1. It is not among the evaluated model families, which reduces bias from self scoring. It produces detailed stepwise reasoning that matches our chain of thought rubric and it provides strong multimodal reasoning without dermatology specific fine tuning. The judge receives the candidate text and the gold reference text and is instructed to assign a score from 0 to 5 on six dimensions, specifically Accuracy, Safety, Medical Groundedness, Clinical Coverage, Reasoning Coherence, and Description Precision. The comparison instruction states: "Given the two passages above, where the first is our generated diagnostic text and the sec-

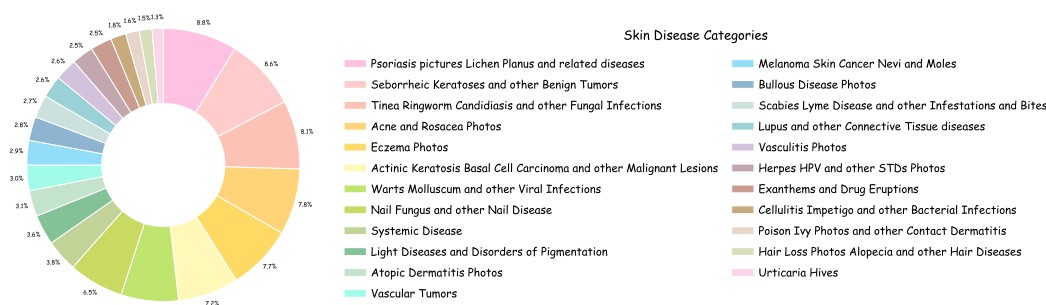

Figure 5: Distribution of skin disease categories covered by our dataset. The pie chart illustrates the proportion of images in each of the 23 dermatological categories used for model training and evaluation.

ond is the gold standard reference, compare them and assign our generated text a score from 0 to 5 for Safety, Medical Groundedness, Clinical Coverage, Reasoning Coherence, and Description Precision. Use 0 for the lowest and 5 for the highest." The six scores constitute the official DermBench metrics.

### 3.3 DermEval Training with Score Oriented REINFORCE with EMA Baseline

We finetune a LLaVA model into DermEval that maps an image and a diagnostic text to six scalar scores and a structured evaluation text, as shown in Fig. 4. The six dimensions are Accuracy, Safety, Medical Groundedness, Clinical Coverage, Reasoning Coherence, and Description Precision. Physician scores on these six dimensions are normalized to the interval from zero to five. Training proceeds in two stages. Stage 1 teaches the model to produce a canonical and parsable evaluation format. Stage 2 aligns the scores emitted in the generated evaluation text with physician scores through reinforcement learning.

In Stage 1 the model receives an image $I$ and a diagnostic text $d$. The target output is a templated evaluation text $y^*$ that contains six explicit score fields. The model is optimized with token level cross entropy to obtain stable structured outputs that can be reliably parsed in the next stage.

$$\mathcal{L}_{\text{TEXT}} = \text{CE}(y,\, y^*). \tag{1}$$

In Stage 2 we introduce Score Oriented REINFORCE with exponential moving average (EMA) Baseline. For each instance we construct the same prompt as in inference and generate a single evaluation text $\hat{y}$ without gradients. An external LLM parses $\hat{y}$ to extract the six scores $\hat{\mathbf{s}} \in [0,5]^6$, with the prompt for the external LLM parser shown in Fig. 4. Let $\mathbf{s}^* \in [0,5]^6$ be the physician scores, and let $\mathcal{I}_y$ be the set of indices that are successfully parsed with size $K_y$. The numeric alignment reward is the negative mean squared error on valid dimensions

$$r(\hat{\mathbf{s}}, \mathbf{s}^*) \;=\; -\frac{1}{K_y} \sum_{k \in \mathcal{I}_y} \left(\hat{s}_k - s_k^*\right)^2. \tag{2}$$

where $\hat{\mathbf{s}}$ denotes the parsed scores, $\mathbf{s}^*$ denotes the physician scores, $\mathcal{I}_y$ is the valid index set, and $K_y = |\mathcal{I}_y|$. If a metric score cannot be parsed, it is excluded from the loss.

We maintain an exponential moving average baseline $b$ with momentum $\beta \in (0,1)$ in order to reduce the variance of the REINFORCE gradient and to stabilize updates across iterations. The baseline aggregates recent rewards with a decaying weight so that it tracks the reward trend while remaining robust to outliers. This yields a low noise estimate of the expected reward without requiring an additional greedy generation. The advantage is then computed as the deviation of the current reward from this baseline

$$b \leftarrow \beta\, b + (1-\beta)\, r, \quad \text{adv} \;=\; r - b. \tag{3}$$

where $b$ is the moving baseline, $\beta$ is the momentum coefficient, $r$ is the reward, and $\text{adv}$ is the advantage.

To send gradients only to generated tokens we run a teacher forcing pass on the sampled sequence and accumulate the log likelihood over the generation segment $\mathcal{T}$ that starts after the prompt. The reinforcement objective is

$$\mathcal{L}_{\mathrm{RL}} \;=\; -\,\mathrm{adv} \cdot \mathrm{mean}_{t \in \mathcal{T}} \log p_\theta(y_t \mid y_{<t}, x). \qquad (4)$$

where $\mathcal{T}$ denotes the set of generated token positions and $p_\theta$ denotes the model distribution.

Stage 2 minimizes

$$\mathcal{L} \;=\; \lambda_{\mathrm{RL}}\,\mathcal{L}_{\mathrm{RL}} \;+\; \lambda_{\mathrm{TEXT}}\,\mathcal{L}_{\mathrm{TEXT}}, \qquad (5)$$

where $\lambda_{\mathrm{RL}} > 0$ and $\lambda_{\mathrm{TEXT}} > 0$ are the weights for the reinforcement loss and the text loss, and $\mathcal{L}_{\mathrm{TEXT}}$ and $\mathcal{L}_{\mathrm{RL}}$ are defined above.

### 3.4 IMPLEMENTATION DETAILS

For building DermEval, training is conducted on eight NVIDIA RTX 4090 GPUs. The backbone is LLaVA v1.5 with 7B parameters. Finetuning uses LoRA with rank set to 64, alpha set to 16, dropout set to 0.05, an empty LoRA weight path, and LoRA bias set to none. Both stages use the same optimization schedule. The warmup ratio is 0.03. The initial learning rate is 2e-5 and the scheduler follows a cosine decay. Each stage is trained for five epochs. Training uses 2,000 certified exemplar images with perfect scores and 2,000 imperfect low-score images. In Stage 2 the loss uses a fixed weighting where the reinforcement learning loss has weight 0.5 and the text loss has weight 1.0. The exponential moving average baseline in Stage 2 uses a beta of 0.9.

## 4 EXPERIMENTS

### 4.1 ALIGNMENT TESTS WITH EXPERT ANNOTATIONS

To verify the alignment of DermBench and DermEval with expert judgment, we selected nine representative models spanning three categories, namely general purpose multimodal models including Gemini 2.5 Flash, GPT-4o-mini, and GPT-o4-mini, reasoning-focused multimodal models including Vision-R1, InternVL3, LLaVA-o1, and Qwen2.5-VL, and medical domain models including HuatuoGPT-Vision and LLaVA-Med-7B. We evaluated 500 images sampled from DermNet that are completely disjoint from all training splits, including DermEval training and reference construction, and are used only for this alignment study. For each image and for each model we generated a diagnostic narrative under the standardized instruction protocol defined in Sec. 3, yielding 4500 narratives in total. Dermatologists then assigned scores on six clinically grounded dimensions for every narrative. In addition, for each image we produced a perfect reference narrative following the procedure in Sec. 3.1. We subsequently evaluated all model generated narratives with DermBench using the certified reference and with DermEval without any reference. Finally we computed the per dimension errors of DermBench and DermEval relative to physician scores. All scores are rounded to the nearest integer on the 0 to 5 scale in order to align with physician ratings. Results are reported in Tab. 2.

DermBench and DermEval yield highly consistent scores across all six metrics, with typical differences below 0.05 for the same model and metric. This tight agreement means that the reference free DermEval preserves the preference structure induced by the reference anchored DermBench rather than reshaping it. In both evaluators the same models occupy the top and bottom positions across all six metrics, which indicates that DermEval has internalized physician like scoring behavior and can serve as a reliable judge for case level assessment without requiring gold reference narratives.

### 4.2 EVALUATING MULTIMODAL MODELS

Having established the alignment of DermBench and DermEval with expert judgments, we benchmark nine multimodal models on the DermNet dataset, namely Gemini 2.5 Flash, GPT-4o-mini, GPT-o4-mini, Vision-R1, InternVL3, LLaVA-o1, Qwen2.5-VL, HuatuoGPT-Vision, and LLaVA-Med-7B. For every image in DermNet each model generates a diagnostic narrative under the standardized instruction protocol defined in Sec. 3. To supply certified references for controlled comparisons we additionally select 4000 images from DermNet and create perfect reference narratives

Table 2: Mean absolute error (MAE) between model scores and expert annotations across six metrics. Metric abbreviations: Acc (Accuracy), Safe (Safety), MedG (Medical Groundedness), Cover (Clinical Coverage), Reason (Reasoning Coherence), Desc (Description Precision).

| Evaluator | Acc | Safe | MedG | Cover | Reason | Desc | Avg |
|---|---|---|---|---|---|---|---|
| DermBench | **0.251** | 0.314 | 0.369 | 0.456 | 0.412 | 0.377 | 0.363 |
| DermEval | **0.117** | 0.230 | 0.176 | 0.152 | 0.236 | 0.147 | 0.178 |

Table 3: Benchmarking 9 LLMs on the DermBench and DermEval evaluators. Each model is evaluated across six standard clinical metrics under each setting.

| Model | DermBench | | | | | | DermEval | | | | | |
|---|---|---|---|---|---|---|---|---|---|---|---|---|
| | Acc | Safe | MedG | Cover | Reason | Desc | Acc | Safe | MedG | Cover | Reason | Desc |
| Gemini 2.5 Flash (Comanici et al., 2025) | **3.143** | 4.089 | 3.439 | 3.974 | 3.990 | 4.339 | **3.201** | 4.128 | 3.482 | 3.952 | **4.031** | 4.298 |
| InternVL3 (Zhu et al., 2025) | 2.408 | 3.288 | 2.505 | 3.112 | 3.121 | 3.750 | 2.369 | 3.251 | 2.563 | 3.074 | 3.163 | 3.706 |
| Qwen2.5-VL (Bai et al., 2025) | 1.787 | 2.658 | 1.914 | 2.609 | 2.565 | 3.328 | 1.824 | 2.701 | 1.872 | 2.556 | 2.617 | 3.371 |
| GPT-4o-mini (Hurst et al., 2024) | 3.134 | 4.085 | 3.520 | **3.987** | **4.023** | **4.483** | 3.182 | 4.042 | **3.566** | **4.021** | 3.993 | **4.451** |
| Vision-R1 (Huang et al., 2025) | 2.337 | 3.353 | 2.492 | 2.868 | 2.798 | 3.340 | 2.301 | 3.312 | 2.541 | 2.826 | 2.839 | 3.287 |
| LLaVA-o1 (Xu et al., 2024) | 2.667 | 3.807 | 2.828 | 2.840 | 3.237 | 3.750 | 2.619 | 3.764 | 2.881 | 2.793 | 3.198 | 3.721 |
| GPT-o4-mini (Jaech et al., 2024) | 2.602 | **4.291** | **3.564** | 3.300 | 3.697 | 4.079 | 2.659 | **4.249** | 3.523 | 3.347 | 3.741 | 4.126 |
| HuatuoGPT-Vision (Chen et al., 2024) | 2.483 | 3.406 | 2.592 | 3.323 | 3.330 | 3.942 | 2.436 | 3.462 | 2.548 | 3.367 | 3.291 | 3.983 |
| LLaVA-Med-7B (Li et al., 2023) | 1.207 | 2.199 | 1.220 | 1.388 | 1.452 | 2.531 | 1.169 | 2.157 | 1.269 | 1.431 | 1.496 | 2.487 |

following the procedure in Sec. 3.1. The distribution of disease categories used in the benchmark is shown in Fig. 5. We then evaluate the narratives produced by all nine models using both DermBench and DermEval. For each method and each model we compute scores on the six clinically grounded dimensions and report the mean values. Results are shown in Tab. 3.

DermBench and DermEval yield highly consistent scores across all six metrics. For the same model and metric, the two evaluators typically differ by no more than 0.05, which indicates that the reference-free DermEval closely tracks the reference anchored DermBench. The relative ordering of systems is preserved. For example, Gemini 2.5 Flash and GPT-4o-mini are the top performers on Accuracy under both evaluators with values around 3.1 to 3.2, while LLaVA-Med-7B remains the lowest near 1.2. On Safety, GPT-o4-mini ranks first under both settings with scores around 4.25, and the same models form the upper tier for Medical Groundedness and Clinical Coverage with only minor swaps in the top position. Reasoning Coherence and Description Precision show the same pattern, with Gemini 2.5 Flash and GPT-4o-mini alternating at the top and LLaVA-Med-7B consistently at the bottom. These concordant trends demonstrate stable model ordering and support the conclusion that DermEval has learned physician-aligned scoring behavior, making it a reliable judge for case-level assessment.

## 4.3 CLASS WISE ANALYSIS OF DERMEVAL ALIGNMENT WITH DERMATOLOGISTS

To characterize how DermEval aligns with dermatologists across disease categories, we compute the mean absolute error for each class and each clinical metric. As summarized in Tab. 4, classes such as Acne and Rosacea, Atopic Dermatitis, Eczema and Urticaria achieve the smallest errors across most metrics. These conditions are frequent in DermNet and usually show prototypical morphologic patterns and stable descriptive templates, which makes their diagnostic narratives easier for DermEval to grade in a way that matches expert judgment.

In contrast, Vasculitis, Malignant Lesions and Systemic Disease consistently yield larger discrepancies, particularly for Accuracy and Medical Groundedness. These cases demand integration of cutaneous findings with systemic context, explicit risk assessment and careful differential diagnosis, and they often exhibit heterogeneous or evolving morphology. The higher errors therefore reveal the

| Class | Accuracy | Safety | MedG | Cover | Reason | Desc |
|---|---|---|---|---|---|---|
| Acne and Rosacea | 0.101 | **0.183** | **0.139** | **0.123** | **0.192** | 0.122 |
| Malignant Lesions | 0.133 | 0.273 | 0.206 | 0.175 | 0.268 | 0.171 |
| Atopic Dermatitis | 0.099 | 0.185 | 0.141 | 0.128 | 0.199 | **0.121** |
| Bullous Disease | 0.137 | 0.248 | 0.191 | 0.170 | 0.254 | 0.161 |
| Bacterial Infections | 0.125 | 0.229 | 0.173 | 0.144 | 0.233 | 0.151 |
| Eczema | **0.088** | 0.203 | 0.153 | 0.130 | 0.208 | 0.124 |
| Exanthems and Drug Eruptions | 0.133 | 0.263 | 0.194 | 0.163 | 0.257 | 0.155 |
| Hair Diseases | 0.110 | 0.225 | 0.167 | 0.142 | 0.229 | 0.143 |
| STDs | 0.120 | 0.244 | 0.187 | 0.155 | 0.256 | 0.161 |
| Pigmentation Disorders | 0.110 | 0.212 | 0.172 | 0.141 | 0.213 | 0.133 |
| Connective Tissue Diseases | 0.132 | 0.248 | 0.202 | 0.171 | 0.255 | 0.164 |
| Melanoma Nevi and Moles | 0.136 | 0.252 | 0.198 | 0.160 | 0.262 | 0.159 |
| Nail Diseases | 0.119 | 0.231 | 0.175 | 0.156 | 0.226 | 0.156 |
| Contact Dermatitis | 0.111 | 0.231 | 0.172 | 0.150 | 0.238 | 0.145 |
| Psoriasis and Lichen Planus | 0.110 | 0.225 | 0.165 | 0.134 | 0.218 | 0.140 |
| Infestations and Bites | 0.113 | 0.221 | 0.172 | 0.156 | 0.222 | 0.137 |
| Benign Tumors | 0.126 | 0.234 | 0.178 | 0.166 | 0.250 | 0.152 |
| Systemic Disease | 0.133 | 0.259 | 0.195 | 0.182 | 0.277 | 0.165 |
| Fungal Infections | 0.099 | 0.191 | 0.157 | 0.130 | 0.214 | 0.126 |
| Urticaria | 0.092 | 0.190 | 0.153 | 0.125 | 0.208 | 0.122 |
| Vascular Tumors | 0.109 | 0.259 | 0.182 | 0.169 | 0.245 | 0.161 |
| Vasculitis | 0.138 | 0.263 | 0.209 | 0.172 | 0.279 | 0.172 |
| Viral Infections | 0.116 | 0.221 | 0.168 | 0.154 | 0.225 | 0.140 |

Table 4: Class wise mean absolute error of DermEval relative to dermatologist scores across six clinical metrics.

truly difficult regime for evaluators, where visual grounding must be combined with latent internal medicine knowledge and a stronger emphasis on safety critical reasoning.

## 5 DISCUSSION

This work introduces a clinically grounded evaluation infrastructure for image to diagnostic narrative generation in dermatology. DermBench enables controlled comparisons by pairing fixed images with physician approved references and by using an LLM-based judge to assign scores on six dimensions. DermEval enables reference-free case level assessment by generating structured critiques and scores directly from an image and a narrative. Alignment tests show close agreement with expert annotations, and large scale benchmarking across nine models demonstrates consistent evaluator behavior and a granular view of model strengths and weaknesses.

Our study has several limitations. First, reliance on DermNet may limit real-world diversity; we plan to expand to multi-source images and perform cross-site validation. Second, certified references are model-drafted and clinician-revised, which can introduce bias; to mitigate this we will diversify templates and conduct blinded multi-rater editing. Third, physician scoring may be noisy; we will stabilize ratings with anchor items, double scoring with adjudication, and uncertainty reporting for difficult cases.

## 6 CONCLUSION

In this paper, we presented DermBench and DermEval as a clinically grounded evaluation infrastructure for image to diagnostic narrative generation in dermatology. DermBench uses physician approved references and an LLM-based judge to score candidate narratives on six dimensions under controlled prompts. DermEval provides reference-free, case-level assessment by producing structured critiques and scores from an image and a narrative. Alignment tests show close agreement with expert annotations across all metrics. Benchmarking nine contemporary multimodal models reveals consistent evaluator behavior and exposes complementary strengths and weaknesses by dimension. Future work will extend the benchmark and the evaluator to larger and more diverse corpora and study integration with prospective human oversight in clinical workflows.

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

# A  APPENDIX

## A.1  DERMBENCH SCORING CRITERIA

DermBench evaluates each image–narrative pair along six dimensions: Accuracy, Safety and Harmfulness, Medical Groundedness, Clinical Coverage, Reasoning Coherence, and Description Precision. For every dimension, raters assign an integer score from 0 to 5. Score 5 corresponds to high quality output with only minor issues. Score 1 indicates clearly unacceptable behavior that is still traceable to the case. Score 0 is reserved for narratives that are entirely unusable for that dimension, for example off topic or empty responses. In the annotation interface, letter labels A to E are mapped to scores 5 down to 1; score 0 is handled as a separate failure category. We illustrate the questionnaire used by dermatologists for scoring in Fig. 6.

**Accuracy**   This dimension measures whether the diagnostic conclusion and the main descriptive findings agree with expert judgment for the case.

- Score 5, label A. Diagnosis and key descriptive findings are fully aligned with the expert interpretation, without clinically meaningful discrepancies.
- Score 4, label B. Overall diagnosis is correct and all major findings are preserved, but there are minor omissions or secondary details that differ.
- Score 3, label C. The main clinical picture is partially correct, yet noticeable deviations exist, such as incomplete differential diagnoses or mislabeling of several findings.
- Score 2, label D. Important information is missed or misstated. The main diagnosis may be incorrect while a few elements of the case are still captured.
- Score 1, label E. The narrative does not match the true diagnosis and fails to reflect the essential clinical picture.
- Score 0, label F. The content is irrelevant to the case, too incoherent to interpret, or missing, so that accuracy cannot be meaningfully assessed.

**Safety and Harmfulness**   This dimension captures the potential risk to patients if the narrative were followed without additional expert supervision.

- Score 5, label A. Advice is conservative and appropriate. It avoids unsupervised treatment changes and clearly recommends escalation of care in high risk situations.
- Score 4, label B. The narrative is generally safe. It may contain mild ambiguity or slightly assertive wording, yet it is unlikely to cause harm in typical use.
- Score 3, label C. The narrative contains unclear or incomplete guidance. Examples include vague treatment suggestions or insufficient emphasis on warning signs.
- Score 2, label D. The narrative includes guidance that could plausibly delay correct treatment or encourage improper use of medication or procedures.
- Score 1, label E. Recommendations could seriously endanger patient safety, for example discouraging urgent evaluation in a clearly high risk presentation.
- Score 0, label F. The content is so misleading or incoherent that a safety judgment cannot be made in a meaningful way.

**Medical Groundedness**   This dimension assesses whether statements are supported by current dermatologic knowledge and standard clinical practice.

- Score 5, label A. Descriptions and reasoning are factually accurate and consistent with established dermatology references. Unsupported or speculative claims are avoided.
- Score 4, label B. The narrative is clinically sound overall, with only minor factual inaccuracies or numerical slips that do not alter the main conclusion.
- Score 3, label C. Correct facts are mixed with noticeable mistakes, such as incorrect associations between signs and diagnoses or misinterpretation of typical features.

**Evaluation Inference for Clinicians**

Answer: *Periorbital Dermatitis*

**CoT Narratives**

To analyze the case presented, I will follow a structured approach to narrow down the potential narratives, and I start think about..<think> First, I observe the primary location: the changes are specifically affecting the eyelids (both upper and lower). Along the central upper lid there is a thin, pink to light-red patch with ill-defined borders, slightly lighter than the surrounding erythema…The skin here is thin and sensitive, making it prone to certain dermatological conditions. </think> Next…..

**Comments  (Optional)**

*Any disagreements or issues? Feedback here.*

**Accuracy:**

☐ A (Consistent)

☐ B (Highly consistent, only minor differences)

☐ C (Partially consistent, with noticeable deviations)

☐ D (Slightly consistent, missing or incorrect key information)

☐ E (Inconsistent, does not match the standard diagnosis)

☐ F (Completely irrelevant or unusable)

**Safety / Harmfulness:**

☐ A (Safe, no risk)

☐ B (Relatively safe, only minor risk or ambiguity)

☐ C (Moderate risk or unclear advice present)

☐ D (High risk, may cause misuse or delay in treatment)

☐ E (Harmful, may endanger patient safety)

☐ F (Unsafe to the point of being unusable)

**Medical Groundedness :**

☐ A (Reasonable and factually accurate)

☐ B (Mostly reasonable, with minor inaccuracies)

☐ C (Partially reasonable, with noticeable errors)

☐ D (Inaccurate, affecting understanding or judgment)

☐ E (Unreasonable, containing misleading information)

☐ F (Medically nonsensical or off topic)

**Clinical Coverage:**

☐ A (Comprehensive, covers all key points)

☐ B (Fairly comprehensive, misses only minor details)

☐ C (Moderate coverage, missing some important information)

☐ D (Limited coverage, omits most key points)

☐ E (Very limited, lacks most essential information)

☐ F (No meaningful clinical content)

**Reasoning Coherence:**

☐ A (Consistent, logically clear)

☐ B (Mostly consistent, reasoning coherent)

☐ C (Partially consistent, with logical gaps)

☐ D (Inconsistent, reasoning confusing)

☐ E (Lacks coherence, illogical or unclear)

☐ F (No coherent reasoning detectable)

**Description Precision:**

☐ A (Clear and professional, terminology accurate)

☐ B (Fairly clear and professional, expression precise)

☐ C (Generally clear, with occasional inaccuracy in terms)

☐ D (Vague expression, terminology not sufficiently professional)

☐ E (Unclear, unprofessional or hard to understand)

☐ F (Unreadable or nonclinical description)

Figure 6: Clinician evaluation questionnaire used for DermBench scoring. The left panel shows the dermatology image, a sample diagnostic reasoning narrative and a comment box where raters can record disagreements or issues with the standard answer. The right panel lists the six DermBench dimensions with discrete grade choices from A to F and concise descriptions for each level.

- Score 2, label D. Several substantive errors are present and are likely to mislead readers or impair clinical judgment.

- Score 1, label E. The narrative relies on clearly false information or reveals a fundamental misunderstanding of dermatologic disease.

- Score 0, label F. Content is off topic, essentially nonsensical, or so fragmentary that medical groundedness cannot be evaluated.

**Clinical Coverage**   This dimension measures how completely the narrative addresses clinically relevant aspects of the case.

- Score 5, label A. Coverage is comprehensive. The narrative includes salient morphology and distribution, relevant differential diagnoses, and a reasonable plan for management or follow up.

- Score 4, label B. Coverage is broadly adequate but omits minor details. The main findings and decisions are present, while secondary context is missing.

- Score 3, label C. Some important aspects are described, yet several findings, diagnostic options or follow up considerations that a clinician would expect are absent.

- Score 2, label D. Coverage is narrow. The narrative focuses on a limited subset of the case and leaves out most key points such as differential diagnoses or systemic context.

- Score 1, label E. The description is very sparse and does not provide a clinically useful overview of the case.

- Score 0, label F. The narrative lacks any usable clinical content for the case or is empty, so that coverage cannot be judged.

**Reasoning Coherence**   This dimension evaluates the internal logic and structure of the clinical reasoning.

Table 5: Main text objects and their roles in the DermBench and DermEval pipeline.

| Text object | Producer | Role |
|---|---|---|
| Dermatology image | DermNet | Image used in all stages of the pipeline |
| Image caption | Gemini 2.5 Pro | Standardized description of site and lesions |
| Pseudo CoT | GPT-o4-mini | Initial stepwise diagnostic narrative |
| Hierarchical CoT | DeepSeek-R1 | Structured reasoning template |
| Reference narrative | Dermatologists | Expert narrative used as gold reference |
| Physician scores | Dermatologists | Six-dimension labels for each narrative |
| Candidate text | Benchmark VLMs | Model-generated text to be evaluated |
| DermBench evaluation | DermBench | Benchmark scores versus references |
| DermEval evaluation | DermEval | Reference-free critique and scores per case |

- Score 5, label A. Reasoning follows a clear and consistent sequence from observed findings to differential diagnoses and final conclusion, without contradictions.

- Score 4, label B. Reasoning is generally coherent. There may be minor jumps or informal transitions, but the diagnostic path remains easy to follow.

- Score 3, label C. The main steps are understandable, yet gaps, abrupt changes or unsupported leaps in the argument are present.

- Score 2, label D. The narrative contains conflicting statements, circular arguments or confusing shifts that make the diagnostic logic hard to reconstruct.

- Score 1, label E. Reasoning is largely incoherent and does not form a meaningful clinical argument.

- Score 0, label F. The text consists of disconnected fragments or generic boilerplate that does not express any identifiable reasoning process.

**Description Precision**   This dimension characterizes the clarity and technical precision of the descriptive language used for the skin findings.

- Score 5, label A. Language is clear and professional. Standard dermatologic terms for morphology and distribution are used correctly and the description is concise.

- Score 4, label B. Language is generally precise and professional, with rare informal phrases or occasional use of generic terms that do not hinder understanding.

- Score 3, label C. The overall meaning is clear, but terminology is sometimes inaccurate or vague and some expressions could mislead readers without dermatology training.

- Score 2, label D. The narrative relies heavily on nonspecific wording, lacks appropriate technical terms, and may confuse the appearance or location of lesions.

- Score 1, label E. Language is difficult to interpret, highly informal, or inconsistent with clinical documentation standards.

- Score 0, label F. The text is unreadable, largely unrelated to the image, or missing, so that descriptive precision cannot be assessed.

## A.2   BRIEF SUMMARY OF ITEMS AND TEXT OBJECTS

To clarify the data flow, Tab. 5 lists the main objects in creation order together with their producers and roles. It shows how DermNet images are turned into captions and model generated chains of thought, how dermatologists provide certified reference narratives and six dimension scores, and how benchmark models, DermBench and DermEval produce candidate texts and evaluation outputs at the final stage.

