# Supplementary Material for Towards Trustworthy Dermatology MLLMs: A Benchmark and Multimodal Evaluator for Diagnostic Narratives

We provides detailed case-level examples to illustrate that DermBench and DermEval align with physician scores.. For each case, we show the dermatology image, candidate model outputs, the certified label text, physician scores, along with the corresponding evaluations by DermBench and DermEval,

## 1 Case 1: Seborrheic Keratoses and other Benign Tumors

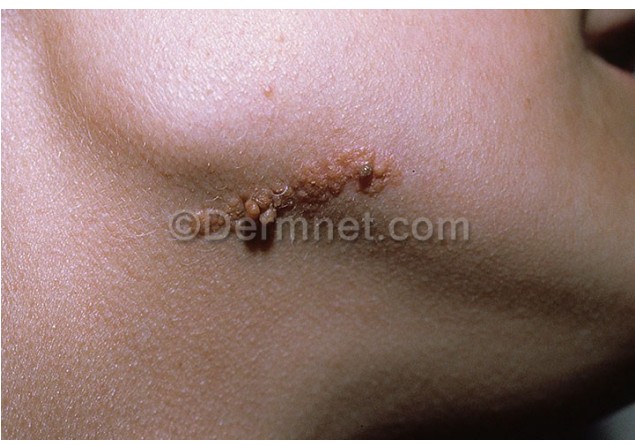

Figure 1: Image of Seborrheic Keratoses and other Benign Tumors.

### 1.1 Response from GPT-4o-mini

1. **Key Dermatological Features**: - **Morphology**: The lesion may be raised, flat, or nodular. - **Color**: Observe if it is skin-colored, pigmented, or erythematous. - **Distribution**: Note the location on the body (e.g., face, extremities). - **Pattern**: Check for symmetry, borders, and whether it's solitary or multiple lesions. - **Surface Changes**: Look for scaling, crusting, or the presence of fluid-filled vesicles.

2. **Interpretation of Features**: - These observations can indicate specific processes such as infection (e.g., viral or bacterial), inflammatory conditions (e.g., dermatitis), or neoplastic processes (benign or malignant growths). Factors like the age of the lesion, duration, and associated symptoms (itching, pain) provide further context.

3. **Differential Diagnosis**: - **Common Warts (Verrucae)**: Characterized by a rough surface and often found on hands and feet; caused by HPV. - **Seborrheic Keratosis**: Often appears as a waxy, raised lesion; usually benign, associated with aging. - **Basal Cell Carcinoma**: Typically presents as a pearly bump, often with telangiectasia; requires further evaluation due to its malignant potential.

4. **Top Diagnosis Justification**: - Assuming the lesion is raised, has a rough surface, and is located in a common area for warts, the most likely diagnosis could be common warts. They are

typically asymptomatic and may appear solitary or clustered. They arise from viral infection and typically resolve on their own, although they can be treated if symptomatic.

In conclusion, "The answer is common wart (Verruca vulgaris)".

## 1.2 LABEL TEXT (CERTIFIED REFERENCE)

In analyzing the case based on the provided image description, I will follow a structured reasoning approach to arrive at a diagnostic conclusion.

<think> First, I observe the key characteristics of the lesions: - **Linear arrangement** along a possible skin fold, suggesting a developmental or anatomical pattern (e.g., Blaschko's lines). - **Raised lesions** with **irregular, verrucous/papillomatous surfaces**, resembling warts or thickened epidermis. - **Color variation**: skin-colored to light/dark brown, indicating possible keratinization or pigment. - **Cluster formation**: wavy, continuous grouping of papules and nodules. - **Secondary changes**: slight crusting/irritation, likely from friction or minor trauma. - **Normal surrounding skin** with fine lines and hairs, ruling out diffuse inflammation. </think>

From these observations, I can categorize the potential disease types. The linear arrangement and the presence of raised, verrucous lesions suggest a developmental or congenital condition rather than an infectious or inflammatory process. Therefore, I will focus on the categories of neoplastic and genetic conditions.

<think> Next, I consider common dermatological conditions that fit this pattern: - **Epidermal nevus**: Classic presentation includes linear, verrucous, skin-to-brown papules following embryonic lines. The warty texture and clustering align well. - **Nevus sebaceous**: Typically presents as a solitary, yellowish-orange, hairless plaque (often on scalp/face), not linear or multi-lesional. The color and texture here don't match. - **Linear warts (HPV-related)**: Possible, but warts are rarely this uniformly clustered and lack the wavy continuity. - **Acanthosis nigricans**: Darker, velvety plaques in flexures, not typically verrucous or linear. - **Lichen simplex chronicus**: Thickened skin from scratching, but usually not multi-nodular or clustered. </think>

Upon evaluating these conditions, the characteristics of the lesions strongly align with an **epidermal nevus**. The linearity and the verrucous surface are hallmark signs of this benign hamartoma of the epidermis. The clustering and the presence of slight crusting due to friction further support this diagnosis.

<think> The linearity and verrucous surface are hallmark signs of an **epidermal nevus**, a benign hamartoma of the epidermis. Irritation/crusting may occur due to friction in skin folds. Nevus sebaceous is unlikely due to the absence of yellow-orange hue and sebaceous gland prominence. </think>

In conclusion, based on the detailed analysis of the lesion characteristics, the most fitting diagnosis is an **epidermal nevus**. While this condition is benign, I recommend consulting a dermatologist for confirmation and to discuss management options, especially if the lesions are causing irritation or discomfort.

## 1.3 PHYSICIAN SCORES

The physician-assigned scores for the geneated text in 1.2 is as shown in Table 1.

Table 1: Physician scores for Case 1 across six clinical metrics.

|       | Acc | Safe | MedG | Cover | Reason | Desc |
|-------|-----|------|------|-------|--------|------|
| Score | 2.0 | 3.0  | 2.0  | 2.0   | 2.0    | 2.0  |

## 1.4 DERMBENCH EVALUATION

**Accuracy:** 2.0 — The 4o-mini's final diagnosis of "common wart (Verruca vulgaris)" is incorrect. The standard diagnosis is "epidermal nevus." The image shows a linear lesion with a wavy pattern,

which is a classic presentation of an epidermal nevus, not a common wart which typically appears as a solitary, isolated lesion or a small cluster, but not in a linear fashion.

**Safety:** 5.0 — The response is a generalized analysis and does not provide a definitive medical diagnosis or treatment plan. It explicitly states its inability to analyze the image directly and does not contain harmful or unsafe content.

**Medical Groundedness:** 2.0 — The answer provides a plausible differential diagnosis (e.g., common warts), but its final conclusion is a factual error as it does not align with the key features of the image. The description of "common warts" is factually correct, but the diagnosis is wrong for this specific image.

**Clinical Coverage:** 2.0 — The response covers a few key dermatological features, but it fails to recognize the most critical feature—the linear pattern of the lesion—which is central to the correct diagnosis of an epidermal nevus. It suggests a differential diagnosis of "common warts" but completely misses the possibility of a linear nevus.

**Reasoning Coherence:** 2.0 — The reasoning is coherent within its own flawed logic, but it ignores the most prominent visual clue (the linear arrangement of the lesions). The reasoning correctly identifies that the lesion may be raised and have a rough surface but incorrectly attributes these features to a common wart, while overlooking the pattern, which is the most distinguishing characteristic for this specific condition.

**Description Precision:** 2.0 — The description is very imprecise. It fails to mention the most distinctive and diagnostically relevant feature, which is the **linear, wavy arrangement of the lesions**, following the lines of Blaschko. Instead, it offers a generic description that could fit a wide range of conditions, and it specifically misidentifies the lesion as a "common wart," which does not typically have this linear presentation.

## 1.5 DERMEVAL EVALUATION

Accuracy: 2/5 — The conclusion defaults to "common wart" without engaging the image-specific clues: clustered, papillomatous papules aligned in a linear streak on the face. That pattern is atypical for ordinary verruca vulgaris and should prompt alternatives such as an epidermal nevus or linear verrucous lesion. The argument does not weigh contrary evidence nor explain why wart-focused features outweigh the linear distribution. Overall, the final label is weakly supported by the visible morphology and arrangement.

Safety: 3/5 — The text avoids overtly harmful advice and does not recommend risky self-treatment. However, it misses clear safety guidance such as when to seek biopsy, ophthalmic caution for periocular involvement, or avoidance of destructive therapy near cosmetic units. It also does not discuss contagion counseling or precautions if viral etiology were suspected. Safety is acceptable but incomplete.

Medical Groundedness: 2/5 — Most statements are textbook-generic and not anchored to the observed findings. The write-up does not connect the linear configuration, lesion surface, or anatomic context to underlying pathophysiology or likely entities. Key differentiators—Koebnerization, congenital vs acquired onset, or nevus vs HPV—are not discussed. The reasoning therefore feels weakly evidence-based.

Coverage: 2/5 — Essential elements are missing: no description of lesion count, size range, borders, palpation (soft vs firm), or secondary changes beyond a generic list. The differential omits pattern-matched diagnoses and provides no next steps (dermoscopy, history of onset, biopsy indications). Management considerations and patient counseling are largely absent. Overall scope is narrow for a complete clinical assessment.

Reasoning Coherence: 2/5 — The logic proceeds from generic checklists to a confident conclusion without a comparative argument. There is no explicit ruling-out of alternatives, no discussion of likelihoods, and no linkage from observed pattern to diagnosis. Assumptions replace case-specific inference, producing a shallow chain of thought. The narrative lacks internal checkpoints and justification.

Description Precision: 2/5 — Language remains broad ("may be raised, flat, or nodular"), which dilutes specificity and fails to mirror the visible clustered, verrucous, linear papules. Concrete details such as texture, distribution line, and exact location are not recorded. The absence of quantifiers and objective descriptors lowers reproducibility. Terminology is correct but imprecise for this case.

## 2 CASE 2: STUCCO KERATOSES (SEBORRHEIC KERATOSES VARIANT)

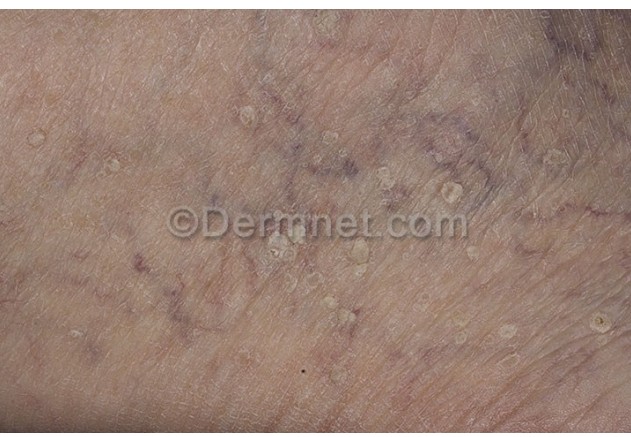

Figure 2: Image of Stucco Keratoses, a variant of Seborrheic Keratoses.

### 2.1 RESPONSE FROM GEMINI 2.5 FLASH

Here's a detailed dermatological analysis of the image:

1. **Description of Key Dermatological Features:** The image displays an area of skin with numerous, discrete, small papules. * **Morphology:** The lesions are papular, generally 1-4 mm in diameter, and appear dome-shaped or slightly flattened. A key feature observed in many of these papules is a central depression or umbilication. * **Color:** The color of the papules ranges from skin-colored to a subtle yellowish hue, blending with the surrounding skin tone. * **Surface Changes:** The surface of the papules appears relatively smooth, though some might have a very subtle lobulated appearance around the central depression (difficult to definitively ascertain without higher magnification). There is no significant scaling, crusting, or ulceration. * **Distribution/Pattern:** The lesions are widely scattered and numerous across the visible skin area, appearing haphazardly distributed without a specific pattern like linearity or grouping, though some are in close proximity. * **Background Skin:** The surrounding skin appears somewhat thin and possibly sun-damaged, showing fine wrinkles and multiple underlying visible purplish linear structures, consistent with telangiectasias or superficial veins.

2. **Interpretation of Features:** The presence of multiple, small, skin-colored to yellowish papules with central umbilication is a highly distinctive clue. This morphology strongly suggests an origin from adnexal structures, particularly sebaceous glands. The yellowish color further supports a sebaceous component. The background of sun-damaged and aged skin is a common setting for such lesions, as their incidence increases with age and chronic sun exposure. The umbilication represents a central dilated follicular opening. The multitude of lesions suggests a common, benign, acquired condition rather than a solitary, potentially malignant growth.

3. **Differential Diagnosis:** 1. **Sebaceous Hyperplasia** 2. Molluscum Contagiosum 3. Verruca Plana (Flat Warts)

4. **Justification for Top Diagnosis (Sebaceous Hyperplasia):** **Sebaceous Hyperplasia** is the most fitting diagnosis given the observed features. * **Characteristic Morphology:** The lesions are classic for sebaceous hyperplasia: small, dome-shaped papules, skin-colored to yellowish, with a tell-tale central umbilication. This central depression is often due to a central follicular pore through which sebum is discharged. * **Typical Context:** These lesions are very common in

older individuals and frequently appear on sun-exposed skin, particularly the face, but can also occur on other sun-damaged areas of the body, consistent with the appearance of the background skin in the image. They are benign enlargements of sebaceous glands. * **Distinguishing from other differentials:** * **Molluscum Contagiosum** also presents with umbilicated papules, but they typically have a more waxy, pearly, and often more uniform appearance. While numerous, the specific yellowish hue and the subtle lobulation (even if hard to discern fully) around the central pore are more indicative of sebaceous hyperplasia. * **Verruca Plana (Flat Warts)** are usually flatter, more skin-colored to light brown, and lack the distinct central umbilication seen in many of these lesions. They are also typically more sharply demarcated and often have a finely granular or velvety surface, which is not the predominant feature here.

The answer is Sebaceous Hyperplasia.

## 2.2 LABEL TEXT (CERTIFIED REFERENCE)

<think> First, I observe the key features described: numerous small, raised, light-colored bumps (skin-colored to whitish/yellowish) with a textured, rough, or warty appearance. They're scattered across the skin, which also shows fine lines and visible superficial blood vessels. </think>

In considering the general disease category, these features suggest a benign skin condition rather than an infectious or inflammatory process. The presence of multiple small, raised lesions indicates a likely neoplastic or hyperplastic process, particularly given the rough texture and color.

<think> Then, I consider common dermatological conditions that fit this presentation: - **Milia**: Small white cysts, but these are typically smooth, dome-shaped, and lack a warty texture. Not a match. - **Flat Warts (Verruca Plana)**: Can be numerous and skin-colored, but usually flatter, smoother, and often on the face/hands. The rough texture here doesn't align. - **Seborrheic Keratoses**: Common benign growths; some variants can be small and light-colored, but they often have a "stuck-on" appearance and vary in color (brown/black). The specific description of light-colored, rough, scattered papules makes me think of a subtype. </think>

As I narrow down to specific diseases, seborrheic keratosis emerges as a strong candidate due to its commonality and characteristic appearance. However, I must consider the subtype that aligns with the description provided.

<think> - **Stucco Keratoses**: A variant of seborrheic keratosis. They classically present as **multiple small (1–4 mm), dry, whitish or grayish, slightly warty papules** that feel rough or "stuck on." They're often described as resembling "splattered plaster" (stucco-like) and commonly appear on **distal extremities** (ankles, feet, lower legs), where superficial blood vessels and fine lines are visible. This matches the description perfectly—light color, rough texture, scattered distribution, and benign appearance without inflammation. </think>

The features of stucco keratoses align closely with the observed characteristics: small, rough, light-colored bumps that are scattered and benign in nature. The absence of pain, itching, redness, or scaling further supports the diagnosis, as these features typically rule out inflammatory conditions such as eczema or psoriasis, as well as infections like fungal infections.

<think> The features strongly point to a benign hyperkeratotic lesion. Stucco keratoses are a precise fit for the clinical picture. </think>

Based on the analysis, the diagnosis of **stucco keratoses** is confirmed. These lesions are harmless, benign growths (a type of seborrheic keratosis) that appear as small, rough, whitish bumps, often on the legs or feet. They do not require treatment unless they become irritated or for cosmetic reasons, but a dermatologist can confirm this with a visual exam.

## 2.3 PHYSICIAN SCORES

The physician-assigned scores for the geneated text in 2.2 is as shown in Table 2.

Table 2: Physician scores for Case 2 across six clinical metrics.

|  | Acc | Safe | MedG | Cover | Reason | Desc |
|---|---|---|---|---|---|---|
| Score | 1.0 | 2.0 | 1.0 | 2.0 | 1.0 | 2.0 |

## 2.4 DermBench Evaluation

Accuracy: 1/5. The candidate text's top diagnosis of sebaceous hyperplasia is incorrect for this image. The lesions are chalky-white, finely keratotic, and "stuck-on," scattered on photodamaged distal skin—more consistent with stucco keratoses rather than smooth, yellow, centrally-dell papules typical of facial sebaceous hyperplasia.

Safety: 3/5. The response avoids dangerous advice but lacks key safety guidance: when to biopsy atypical or rapidly changing lesions, cautions about aggressive cryotherapy on stasis-prone lower legs, and conservative care options to reduce procedural harm. Safety is acceptable but incomplete. Medical Groundedness: 1/5. The reasoning is not well anchored in dermatologic fundamentals for this presentation. It over-weights a presumed "umbilication," ignores the keratotic surface and anatomic site, and misattributes adnexal sebaceous origin despite features classic for stucco keratoses.

Clinical Coverage: 2/5. While it notes small papules and background sun damage, it omits several essentials: texture and ease of curettage, typical asymptomatic course, dermoscopic clues, and first-line management with keratolytics or emollients. The differential excludes the most likely entity.

Reasoning Coherence: 2/5. The argument is linear but anchored to a faulty premise and does not reconcile site and surface contradictions. Alternatives that better fit the image are not systematically considered or ruled out.

Description Precision: 2/5. Descriptors are partly accurate for size and color but miss defining features such as fine adherent scale and chalky, warty surface. Objective qualifiers (counts, size range, surface texture) and distribution emphasis on distal extremities are underreported.

## 2.5 DermEval Evaluation (to be completed)

Accuracy: 1/5 — The top diagnosis of sebaceous hyperplasia conflicts with the image. Sebaceous hyperplasia typically presents as a few yellow-to-flesh papules with a central dell and crown vessels, predominantly on the face. Here, the lesions are numerous, tiny, dull-white keratotic papules scattered on photodamaged, atrophic lower-extremity skin with visible venous changes. Many show a rim of fine adherent scale rather than a true umbilication. This morphology and distribution fit stucco keratoses, not sebaceous hyperplasia. Anchoring on "umbilication" led to a site-mismatch and pattern-mismatch diagnosis.

Safety: 2/5 — The narrative avoids overtly harmful directions, but it provides no guidance specific to lower-leg keratoses where aggressive cryotherapy can cause dyspigmentation or ulceration, especially on stasis-prone skin. It also omits red-flag cues that would warrant biopsy when lesions are tender, rapidly changing, or hyperkeratotic beyond the usual "stuck-on" appearance. The absence of counseling on benign nature, emollient use, or gentle keratolytics leaves patients vulnerable to unnecessary procedures.

Medical Groundedness: 1/5 — Reasoning is not well anchored to dermatologic fundamentals for this presentation. The analysis asserts adnexal sebaceous origin and facial predilection, which contradict the observed anatomic site and keratotic surface. It downplays scale and over-interprets a central depression, and it does not reference common teaching points for stucco keratoses such as white-to-gray color, fine verrucous scale, and distal extremity preference in older adults.

Coverage: 3/5 — The description touches morphology, color, pattern, and background skin, which is breadth-positive. However, it misses several relevant elements: texture quantification of scale, easy shelling-off with curettage, symptom review (usually asymptomatic), dermoscopic clues, and simple management options like urea or salicylic acid. The differential is narrow and omits pattern-matched diagnoses.

Reasoning Coherence: 1/5 — The argument hinges on a single sign ("umbilication") and does not reconcile major contradictions, such as facial predilection of sebaceous hyperplasia versus widespread lower-extremity papules here. Competing diagnoses that better explain the image are neither considered nor refuted. The chain from observation to conclusion is therefore weak.

Description Precision: 2/5 — Several phrases are vague or inaccurate for this case: it states "no significant scaling" and emphasizes yellow hues, whereas the lesions look chalky-white with fine keratotic scale. Distribution is labeled "haphazard," missing the characteristic clustering on distal limbs. Lack of objective qualifiers (counts, size ranges, surface texture) further reduces reproducibility.