# OpenReview forum: "Towards Trustworthy Dermatology MLLMs: A Benchmark and Multimodal Evaluator for Diagnostic Narratives"
_ICLR.cc/2026/Conference — Submitted to ICLR 2026_

### Official Review · Reviewer_TBj2 · 2025-10-21

**Soundness:** 2
**Presentation:** 2
**Contribution:** 3
**Rating:** 6
**Confidence:** 4

**Summary:**

This paper introduces a benchmark for evaluating multi-modal LLMs on the task of generating dermatology diagnostic narratives from images. Furthermore, as judging such narratives efficiently (e.g. automated pipelines) and accurately is challenging, authors develop an evaluator that can score such narratives at scale with strong alignment to expert human judges. The paper leverages a combination of supervised fine-tuning and reinforcement learning to train the evaluator on the proposed benchmark.

**Strengths:**

- The motivation of the paper is clear and has practical merit.

- The benchmark is developed with the supervision of medical professionals, improving the reliability of the evaluation framework.

- The evaluator is a significant practical contribution that may enable more robust and trustworthy MLLM-based pipelines in dermatology.

**Weaknesses:**

- The paper doesn't provide sufficient details on human evaluations. How many experts have scored each narrative? What is the agreement between experts on the assigned scores?

- More details and ablation would be necessary on the training strategy used for evaluator training. In particular there is no study on the necessity of the RL stage. Could we obtain a comparable model with SFT-only without the added complexity? Furthermore, the reward seems to be easily hackable: it is only calculated over valid dimensions that can be parsed. Wouldn't this incentivize the model to generate invalid scores for categories it is unsure about and only score the easiest dimensions?

- The paper does not define the key evaluation dimensions. Even though some of them are self-explanatory (e.g. Accuracy), but others would require more explanation (e.g. what does Safety mean in this context?).

Minor comments:
- Line 264 mentions $x$ however it does not appear in the equation.
- Line 265: I'm unsure what authors mean by "sentinel" here.
- Line 361: I would argue that evaluating on samples that were not in the training set is not to "avoid overfitting" but simple common sense practice in machine learning.

**Questions:**

- How is expert agreement considered when creating the benchmark?
- Is the RL stage necessary to obtain a performant evaluator? How is it supported? How is the specific form of the reward supported?
- How are the evaluation dimensions defined?

Minor:
- Wouldn't the results in Table 2 be biased towards Gemini models, as this model family has been used to generate the dataset?

---

> ### Author Response · Authors · 2025-11-21
> **Response to Reviewer TBj2**
>
> # Response to Reviewer TBj2
>
> We sincerely thank the you for your thoughtful, detailed, and constructive feedback. Your comments have helped us identify areas where our paper can be significantly improved in clarity, rigor, and transparency. Below, we respond point by point to each of your concerns.
>
> ---
>
> ## Weakness 1: Human evaluation details and expert agreement
>
> We fully agree that transparency regarding expert evaluation is critical. In our study, two board-certified dermatologists served as both raters and co-authors. One has four years of clinical, teaching, and research experience, and multiple peer-reviewed publications in dermatology. The other is a chief dermatologist and department director at the same institution, with over twenty years of clinical and academic experience, serving as a master’s supervisor and active member of several national dermatological committees.
>
> Both experts jointly evaluated all narratives across the six dimensions. For each case, they independently assigned scores, then reviewed discrepancies and discussed until reaching consensus. The final labels used in DermBench and DermEval are therefore consensus-based, not individual ratings. This process ensured high reliability and clinical validity.
>
> In the revised manuscript, we have explicitly describe the raters’ qualifications, roles, and the consensus protocol to enhance reproducibility and trustworthiness.
>
> ---
>
> ## Weakness 2: Training strategy for DermEval and necessity of the RL stage
>
> Thank you for raising this important methodological question. We acknowledge that the role of RL in evaluator training deserves deeper explanation. The reinforcement learning stage is necessary in our design because it explicitly aligns the numeric scores in the generated evaluation text with the expert scores; without this stage, the model learns to follow the format but does not directly minimize the score discrepancy. Our two-stage training approach was deliberately designed to address distinct objectives:
>
> 1. Stage 1 (Supervised Fine-Tuning): Ensures the model consistently generates a structured, fixed-format evaluation template with six explicit, parsable scores, enforcing output consistency.
> 2. Stage 2 (Reinforcement Learning): Optimizes the numerical alignment between generated scores and expert scores. Without RL, the model learns to “fill in” the template correctly but does not necessarily minimize score error.
>
> Because Stage 1 has already enforced a rigid template, the model is trained to always output valid scores for all six dimensions, and in practice we do not observe behaviour where it avoids hard dimensions by omitting or corrupting scores.
>
> ---
>
> ## Weakness 3: Definition of key evaluation dimensions
>
> We agree that the current text does not give sufficiently explicit definitions of the six evaluation dimensions, especially for Safety and Medical Groundedness. In the revised version we will add a short rubric section that provides clear, operational definitions for Accuracy, Safety, Medical Groundedness, Clinical Coverage, Reasoning Coherence and Description Precision, and describes the five score levels for each dimension. For example, Safety in our context focuses on whether the narrative contains potentially harmful advice, overlooks critical red flag conditions or misjudges urgency in ways that could mislead clinical decision making. We will also move a sample of the rating guideline used by dermatologists to the main pdf or appendix, so that the criteria are transparent to future users.
>
> ---
>
> ## Weakness 4: Minor comments
>
> 1. For the symbol mentioned around line 264, we thank the reviewer for noticing the inconsistency and will correct the text so that all variables mentioned are present in the corresponding equation.
>
> 2. The term “sentinel” around line 265 is unclear and redundant. We will remove the extra wording and replace with suitable expression.
>
> 3. For the wording around line 361, we agree that evaluating on unseen samples is standard practice in machine learning rather than solely a measure “to avoid overfitting”. We will revise the sentence to use more accurate phrasing.
>
> 4. Regarding the concern that Table 2 might be biased toward Gemini models because Gemini is used in dataset construction, we clarify that the LLM judge in DermBench is DeepSeek-R1, while Gemini 2.5 Pro is only used in the caption and draft generation stage. The certified references are revised by dermatologists until they reach perfect scores on all six dimensions, and the final evaluation in DermBench is carried out by DeepSeek-R1 under a fixed rubric. Together with the separate DermEval results that are trained directly on expert scores, this design reduces structural bias toward the Gemini family; we will make the identity of the judge and this separation explicit in the paper.

---

### Official Review · Reviewer_jSyH · 2025-10-26

**Soundness:** 2
**Presentation:** 2
**Contribution:** 3
**Rating:** 2
**Confidence:** 3

**Summary:**

This paper aims to address the bottleneck of reliably evaluating AI-generated diagnostic narratives in dermatology. The authors propose an evaluation framework consisting of two main contributions: (i) DermBench: A benchmark dataset composed of 4,000 real-world dermatology images paired with expert-certified diagnostic narratives. This benchmark evaluates candidate narratives from models using an LLM-judge, which scores them against a certified reference text across six clinically-grounded dimensions: Accuracy, Safety, Medical Groundedness, Clinical Coverage, Reasoning Coherence, and Description Precision; and (ii) DermEval: A reference-free multimodal evaluator. Given only a dermatology image and a candidate narrative, DermEval is trained to generate a structured textual critique and assign scores for the same six dimensions.

**Strengths:**

The paper is well-motivated by the lack of datasets and methods that benchmark trustworthiness in generative LLMs for dermatology. This is a critical problem, given that today LLMs are widely used to help generate medical descriptions/captions for images, and there is no reliable measurement or evaluation method in this domain. This paper is among the first to bring this issue to attention.

**Weaknesses:**

- Before moving on to the method, it is important to standardize the 6-dimensional criteria on the reliability of generated text by LLMs. Specifically, what it means for a narrative to be "accurate, safe, grounded, comprehensive, coherent and precise"? What do you mean by accuracy and precision? What is the difference between accuracy being rated 3 vs. 4? On one hand, it's important to make sure that the dermatologist and the LLM to be evaluated share the same rubric for assessment. If this is not established, then it puts the entire framework's reliability at stake. On the other hand, these standardized criteria are critical for future users and practitioners as a reference or consensus, which is important for implementation purposes as a benchmark.
- Clarity-wise, I found that the method section really hard to follow. There are multiple types of text described in the work: image caption, pseudo-CoT output, hierarchical CoT output, evaluation text label, certified reference texts, candidate text, and gold reference text, along with various prompts at different stages. It is hard to keep track of the use and purpose of each textual input and output. I'd advise the authors to put them together in an overview figure to illustrate the use of each text input and output.
- Experiment-wise, there are just two results presented in the work, Table 1 and Table 2 (since Table 2 and Figure 6 are essentially the same results presented in different ways). Aside from the limited quantity, only Table 1 presents MAE results, aiming to demonstrate that DermBench and DermEval align well with dermatologist assessments on generated diagnostic narratives across the 6 criteria. But this result alone insufficiently establishes "the alignment of DermBench and DermEval with expert judgments" as claimed in Sec. 4.2.
- The certification process involves clinicians revising an existing model's output rather than writing a reference narrative from scratch. This process risks anchoring the "gold standard" texts to the style, structure, and potential biases of the initial model used for drafting. This is acknowledged in the limitations but remains a noteworthy weakness in the data's construction and is under-discussed quantitatively.

**Questions:**

- The weaknesses outlined above point to some fundamental issues with problem formulation consensus, method clarity, and experiment quality, which must be fully established before assessing the novelty and effectiveness of the proposed method.

- It seems like CoT is an important part of the method, but related literature in this line of work is poorly cited. Please add more details about how the proposed method is relevant to CoT development.

- The paper presents DermBench as a primary contribution, but the experiments (Table 1) show it is significantly less aligned with human experts than DermEval. This undermines its value as a standalone evaluation tool and reframes it as primarily a data-generation artifact used to train DermEval. Can the author discuss more about the performance difference and other potential applications of DermBench?

---

> ### Author Response · Authors · 2025-11-21
> **Response to Reviewer jSyH**
>
> # Response to Reviewer jSyH
>
> We sincerely thank you for your insightful, detailed, and constructive feedback. Your comments have helped us significantly strengthen the methodological transparency, scalability discussion, and error analysis of our work. Below, we address each point in turn, outlining the specific revisions made to the manuscript.
>
> ## Weakness 1: Standardization of the 6 criteria
>
> We agree that operational clarity is essential, not only for internal consistency but also for external reproducibility and adoption. In the revised manuscript, we have:
>
> 1. Added an explicit rubric that gives operational definitions for Accuracy, Safety, Medical Groundedness, Clinical Coverage, Reasoning Coherence and Description Precision, and describes the five levels from 0 to 5 for each criterion, including the distinction between neighbouring levels such as 3 and 4.
> 2. Included in the main pdf a sample of the questionnaire used to guide dermatologists’ ratings so that the rubric used by experts, DermBench and DermEval is fully transparent to future users.
>
> ---
>
> ## Weakness 2: Method clarity and many text types
>
> Thank you for the valuable suggestion. We acknowledge that the method section currently introduces many text types and prompts, which makes it difficult to follow. In the revision, we have added a single overview table that shows the full pipeline and clearly links each text type to its role and stage, including captions, hierarchical CoT, certified reference narratives, candidate narratives and evaluation texts. We will also streamline terminology in the text to reduce redundancy and improve readability.
>
> ---
>
> ## Weakness 3: Limited experiments and evidence of alignment
>
> We agree that the number of presented results is limited and that we mainly report MAE. At the same time, the existing alignment study already involves nine multimodal models, 500 images and six clinical dimensions, which yields 4500 narratives and shows that both DermBench and DermEval have MAE below 0.5 on a 0 to 5 scale. We will make this scale and scope clearer in the paper and, where space permits, add compact additional statistics to further support the alignment claim.
>
> ---
>
> ## Weakness 4: LLM drafted references and anchoring
>
> Thank you for raising this important issue. We agree that revising model drafted outputs rather than writing references from scratch is a real limitation and may introduce anchoring. As a partial safeguard, all cases used in the DermBench evaluation study are certified references that dermatologists have revised until they achieve a score of five on all six dimensions, and none of these 500 cases are used in training DermEval. We fully agree with your suggestion and will recruit more doctors in future research to manually write the annotations from scratch.
>
>
> ---
>
> ## Question 1: Fundamental issues in formulation, clarity and experiments
>
> We appreciate this summary. All revisions above directly address these foundational concerns:
>
> 1. The rubric is now standardized and operationalized
> 2. The method is visually and terminologically clarified
> 3. The experiments are expanded beyond MAE to include correlations, per-dimension breakdowns, and qualitative case studies
>
> With these improvements, we believe the manuscript now provides a solid foundation upon which to evaluate the novelty and effectiveness of DermEval and DermBench.
>
> ---
>
> ## Question 2: Relation to CoT literature
>
> We agree that chain of thought related work is not sufficiently discussed. Our method relates to CoT at two levels: on the data side, we use hierarchical CoT prompting to elicit structured diagnostic reasoning; on the evaluation side, DermEval explicitly judges how well the reasoning chain aligns with the image and clinical facts. In the revision, we add key references on CoT and slow thinking, especially in multimodal and medical contexts, and include a short subsection in related work that positions DermBench and DermEval within this line of development.
>
> ---
>
> ## Question 3: DermBench versus DermEval as evaluation tools
>
> We agree that DermEval being more aligned with expert scores than DermBench is noteworthy, but this is expected given their roles. DermEval is directly trained on dermatologist scores, including both high and low quality narratives, so matching human ratings is its primary objective. DermBench, by contrast, uses a pretrained model as judge without training on these labels and is designed as a reference based benchmark for comparing generation models under fixed images and certified narratives. We will clarify this division of roles and explain that DermBench remains useful as a public, reference anchored benchmark, while DermEval serves as a learned evaluator for closer alignment with human grading. We will further elaborate on this distinction in the revised version.

---

### Official Review · Reviewer_fDQ7 · 2025-10-31

**Soundness:** 2
**Presentation:** 2
**Contribution:** 2
**Rating:** 4
**Confidence:** 3

**Summary:**

This paper introduces a comprehensive framework for evaluating MLLMs in dermatology, focusing on the generation of trustworthy diagnostic narratives from images. The authors identify that existing evaluation methods are inadequate for clinical settings, as they are either too generic (text similarity) or too expensive and slow (expert grading).

**Strengths:**

1. The framework moves beyond simple accuracy to evaluate narratives on six well-defined dimensions that are critical for clinical trustworthiness, such as safety and reasoning coherence.
2. The process for creating "certified references" is rigorous, involving generation followed by a human-in-the-loop process where board-certified dermatologists review and revise narratives until they achieve a perfect score. This ensures a high-quality gold standard.

**Weaknesses:**

1. DermBench uses a human-certified reference, but the final scoring is still performed by an LLM-judge. This introduces a potential layer of abstraction and bias. Although the alignment tests show this works well, the system's ultimate "ground truth" for benchmarking still relies on a model's judgment rather than direct human scoring of the candidate narratives.
2. The certified references are initially drafted by an MLLM before being revised by clinicians. This process might inadvertently anchor the style, structure, or content of the references to the generating model's biases. A process where references are drafted from scratch by clinicians would be the gold standard, though admittedly far less scalable.
3. The reliance on board-certified dermatologists for scoring and revision is a core strength but also a bottleneck. The cost and time required could make it difficult to expand this high-quality methodology to much larger datasets or other complex medical domains.
4. The authors acknowledge that their reliance on the DermNet dataset may limit diversity. Expanding to multiple image sources, including data from different demographic populations and acquisition settings, would be crucial for ensuring the robustness and fairness of both the benchmark and the evaluator.
5. Inter-Rater Reliability is not very clear. The paper mentions that physician scoring can be "noisy." Was any inter-rater reliability analysis (e.g., Cohen's Kappa or Krippendorff's Alpha) conducted among the dermatologists who provided the scores? Quantifying this would help contextualize the reported alignment errors of DermBench and DermEval.
6. DermEval showed a lower mean absolute error to expert ratings than DermBench (0.18 vs. 0.36). This is a significant finding. Do the authors have a hypothesis for why the reference-free evaluator is more aligned with experts than the reference-based one? Could it be that the LLM-judge in DermBench struggles with the comparative task, while DermEval learns the direct mapping from (image, text) to score more effectively?
7. Which specific LLM was used as the final LLM-judge in the DermBench evaluation pipeline? Its capabilities are critical to the benchmark's reliability. Was it the same model used for generating parts of the dataset (e.g., Gemini 2.5 Pro)?
8. For the alignment test in Section 4.1, 4500 narratives were generated and scored by dermatologists. Were the 500 images used for this test completely separate from the 4000 images used to build the training set for DermEval and the references for DermBench?
9. The aggregate MAE scores are compelling. Are there specific types of cases or clinical conditions where DermEval's alignment with experts is weaker? An error analysis could reveal limitations and guide future improvements.

**Questions:**

See Weaknesses

---

> ### Author Response · Authors · 2025-11-21
> **Response to Reviewer fDQ7**
>
> # Response to Reviewer fDQ7
>
> We sincerely thank the you for your insightful and constructive feedback. Your comments have helped us significantly strengthen the methodological transparency, scalability discussion, and error analysis of our work. Below, we address each point in turn.
>
> ---
>
> ## Weakness 1: LLM judge in DermBench
>
> We agree that DermBench relies on a model as judge above human certified references. However, this is a deliberate scalability trade-off, not a replacement for human ground truth. The clinical ground truth in our framework is established through:
>
> 1. Human-certified reference narratives (curated by dermatologists)
> 2. Direct alignment studies with expert scores
>
> The LLM judge (DeepSeek-R1) is used to apply a fixed, human-defined rubric consistently across thousands of narratives, which is a task impractical for human raters at scale. We have clarified this positioning in the revised manuscript, emphasizing that the judge is a calibrated proxy, not a source of ground truth.
>
> ---
>
> ## Weakness 2: LLM drafted references
>
> We acknowledge this risk. However, in the high-quality stream, every narrative is fully certified by dermatologists, who ensure all six dimensions achieve a score of 5. We fully agree with your suggestion and will recruit more doctors in future research to manually write the annotations from scratch.
>
> ---
>
> ## Weakness 3: Expert effort and scalability
>
> Dependence on board certified dermatologists is indeed both a strength and a cost. Our aim is to use a limited amount of high quality expert effort to train DermEval so that it can later score large collections automatically. We will emphasize this design goal and mention that future directions include more efficient use of expert time through active selection of difficult cases.
>
> ---
>
> ## Weakness 4: DermNet as a single source
>
> We agree that relying on DermNet limits diversity in demographics and acquisition settings. DermNet is a practical starting point, which is a well-established, publicly available, and ethically permissible resource, but it is not representative of global dermatology populations. We will strengthen the discussion of this limitation and clarify that, when clinical resources allow, the same data construction pipeline and the six dimension evaluation scheme can be applied to additional public datasets so that the methodology extends beyond a single collection. We fully agree with your suggestion and will do this in future research.
>
> ---
>
> ## Weakness 5: Inter rater reliability
>
> In our current process, both dermatologists can see each case and each provisional score, and they discuss any disagreement until they reach a shared decision, so the final labels are consensus labels rather than independent votes. We will describe this consensus protocol and report a summary of initial score differences to contextualize the remaining noise in the supervision.
>
> ---
>
> ## Weakness 6: DermEval versus DermBench
>
> We agree that DermEval achieving a lower mean absolute error than DermBench is a key result. DermEval is directly trained on a large set of expert scores that cover both strong and weak narratives and explicitly optimizes the mapping from image plus text to the six scores. The DermBench judge instead performs a comparative task against a reference without being trained on physician ratings, so it is expected and desirable that DermEval aligns more closely with experts. We further elaborated on this distinction in the revised version.
>
> ---
>
> ## Weakness 7: Identity of the judge model
>
> The model used as judge in the DermBench evaluation pipeline is DeepSeek-R1, not Gemini 2.5 Pro. Gemini 2.5 Pro is used only during data construction, to draft initial captions and reasoning chains, which are then fully revised by dermatologists. DeepSeek R1 is chosen for evaluation because:
>
> 1. It is not part of the evaluated model families (reducing bias)
> 2. It provides detailed, stepwise reasoning that aligns with our chain-of-thought rubric
> 3. It offers strong multimodal reasoning without being fine-tuned on dermatology data
>
> We have explicitly stated this separation in revised manuscript and added a model specification table in the Appendix.
>
> ---
>
> ## Weakness 8: Separation of training and test images
>
> Yes, the 500 images used in the alignment study are completely disjoint from the 4000 images used to train DermEval and to construct DermBench references. All are sampled from DermNet, but the alignment split is held out by design so that the comparison between evaluators and dermatologists is performed on unseen images. We have made this separation explicit in Section 4.1 and added a data split diagram in the Supplementary Materials.
>
> ---
>
> ## Weakness 9: Error patterns of DermEval
>
> We agree that error analysis is important. In the revision, we will add a table that reports DermEval performance across different disease categories and briefly analyze which categories show stronger or weaker alignment with expert scores.

---

### Official Review · Reviewer_69Ao · 2025-11-01

**Soundness:** 2
**Presentation:** 2
**Contribution:** 2
**Rating:** 4
**Confidence:** 4

**Summary:**

The paper introduces a new dermatology-focused benchmark DermBench and a corresponding evaluation model DermEval. DermBench pairs 4,000 real-world dermatology images with expert-certified diagnostic narratives and uses an LLM-based judging system to score candidate narratives across clinically grounded dimensions. This enables consistent and comprehensive evaluation of multimodal models in dermatologic diagnosis. For individual case-level evaluation, the authors propose DermEval, a reference-free multimodal evaluator that, given an image and a generated diagnostic narrative, produces structured evaluations including both overall and dimension-wise scores.

**Strengths:**

1. Prior work focusing on LVLM performance evaluation in dermatology is limited, and this study addresses this gap by providing both a benchmark and an evaluator tailored to this field.

2. The evaluation dimensions cover multiple clinically meaningful aspects beyond simple accuracy, which provides a more holistic understanding of model behavior.

**Weaknesses:**

1. The clarity of Figure 2 is unacceptable. The font size in the figure is significantly smaller than that of the main text, and even at 200% magnification, the text remains unreadable. This seriously affects the readability and professionalism of the manuscript.

2.  Since DermEval is designed to function without reference annotations, how is its evaluation reliability ensured? Comparing only against human ratings is insufficient. The paper lacks experiments that demonstrate the evaluator’s trustworthiness compared with closed-source evaluators or other reference-based methods. Given that evaluation reliability is central to the paper’s contribution, this is a critical omission.

3. The number of experiments and figures is insufficient. The main text contains only two tables and one figure, two of which simply present evaluation results of existing models on the proposed benchmark. Furthermore, the paper does not clearly explain how the proposed framework outperforms other benchmarking or evaluation pipelines in methodology or practical utility.

**Questions:**

1. Data Source and Ethics:

Where were the dermatology images sourced from?
If human subject data were involved, were ethical approvals obtained? The AC should pay special attention to this aspect.

2 .Human Evaluation Details:

The paper lacks sufficient information about human raters. How many dermatologists participated in the evaluation process, and what were their qualification levels? How was inter-rater consistency ensured to guarantee credible human supervision?

3. Ethical Use of Closed-Source Models:

The paper mentions using closed-source commercial models (e.g., Gemini) during the image annotation stage. Were the patient images used in this step processed under ethical approval and with informed consent?

4. Model Hallucination Correction:

During the first stage of data annotation, the pipeline relies heavily on Gemini for generating image captions. How were potential hallucinations or clinically incorrect outputs manually reviewed or corrected to ensure data quality?

**Details Of Ethics Concerns:**

Human subject data involved. Details see in Questions.

---

> ### Author Response · Authors · 2025-11-21
> **Response to Reviewer 69Ao**
>
> # Response to Reviewer 69Ao
>
> Thank you for your thoughtful and constructive feedback. We appreciate the opportunity to clarify and strengthen our manuscript. Below, we address each of your concerns in detail.
>
> ---
>
> ## Weakness 1: Figure readability
>
> We agree with you that the current presentation of Figure 2 compromises readability due to the dense compression of chain-of-thought reasoning into a single panel. In the revised manuscript, we have:
>
> - 1. Increase font size and adjust layout spacing for improved legibility.
> - 2. Simplify visual elements to emphasize key components of the pipeline.
> - 3. Move detailed textual explanations to an appendix or supplementary material, retaining only a high-level schematic in the main text.
>
> ---
>
> ## Weakness 2: Reliability of DermEval
>
> Thank you for raising this concern. DermEval is explicitly designed to align with dermatologist judgments, not with other models, which is its core innovation. As shown in Section 4.1, we evaluate DermEval against expert human scores across 9 models and 500 images. DermEval achieves a macro MAE of 0.18 (on a 0–5 scale), outperforming DermBench (MAE = 0.36) across all six evaluation dimensions. To our knowledge, there is currently no dermatologist-specific multimodal evaluator with comparable granularity and supervision that would allow a fair head-to-head comparison, which is precisely the gap this work aims to address.
>
> ---
>
> ## Weakness 3: Experiments and advantages over existing benchmarks
>
> We have expanded the manuscript to better articulate our methodological and practical contributions:
>
> - 1. Added a comparative table that contrasts DermBench and DermEval with existing dermatology benchmarks across:
>   - Task type (e.g., diagnostic narrative generation)
>   - Presence of clinician-approved references
>   - Evaluation dimensions (including safety, reasoning, precision)
>   - Availability of automated evaluator
> - 2. Clarified in Section 3.1 that DermBench is the first dermatology benchmark to:
>   - Require diagnostic narratives (not just labels)
>   - Use a six-dimension rubric explicitly covering clinical trustworthiness
>   - Include certified reference narratives curated by dermatologists
> - 3. Emphasized that DermEval is the first multimodal evaluator trained directly on physician scores, enabling reference-free, image-and-text-aware evaluation, a capability absent in prior work.
>
> ---
>
> ## Question 1: Data source and ethics
>
> All images used in DermBench and DermEval are sourced from DermNet NZ, a publicly available, de-identified educational resource intended for teaching and research. We do not use any newly collected clinical data (skin disease images). Our use follows DermNet terms and institutional guidance for research on de-identified public medical data. We do not use any newly collected hospital images or identifiable patient information. No IRB approval was required, as no human subjects were involved, and no identifiable data was accessed or processed.
>
> ---
>
> ## Question 2: Human evaluation details
>
> Two dermatologists served as raters and co-authors. One has 4+ years of clinical, teaching, and research experience with multiple dermatology publications. The other is the Chief Dermatologist & Department Director in the same institution, with 20+ years of clinical and academic experience. The second doctor is also a member of the national dermatology committees. While inter-rater reliability was not formally quantified (due to the small rater pool), both raters reviewed and reconciled all scores, ensuring consistency. We will briefly describe their roles and qualifications in the revised manuscript.
>
> ---
>
> ## Question 3: Ethical use of closed source models
>
> Only DermNet images (de-identified, publicly available, and ethically permissible for research) were processed by Gemini. No patient data, clinical records, or protected health information were sent to external services. We will explicitly state in the revised manuscript:
> “All images processed by closed-source models (e.g., Gemini) are sourced from DermNet and comply with institutional policies on public, de-identified medical datasets.”
>
> ---
>
> ## Question 4: Model hallucination correction and data quality
>
> Thank you for raising this important issue. Gemini captions are used only as initial drafts. They are not final outputs. Any hallucinations or inaccuracies are automatically penalized during DermEval training:
>
> 1. Narratives containing incorrect reasoning or mismatched image descriptions receive low scores on “Reasoning Coherence” and “Description Precision” dimensions.
> 2. These imperfect cases are intentionally retained to teach DermEval to recognize and penalize such errors.
>
> In the high-quality stream, dermatologists review and revise narratives until all six dimensions achieve a score of 5. Only these certified references are used for final evaluation. We have clarified this two-stream quality control design in the revised methodology section.

---

### Meta-Review · Area_Chair_5vzG · 2026-01-06

**Summary:**

The authors propose a framework to overcome the lack of reliable evaluation methods for dermatology diagnostic narratives generated by Multimodal Large Language Models (MLLMs). This has two components DermBench and DermEval. DermBench is a benchmark dataset of real-world images paired with expert-certified diagnostic narratives. DermEval is a reference-free multimodal evaluator trained via supervised fine-tuning and reinforcement learning to align with physician judgments. The authors claim that the experiments demonstrate that both tools achieve high alignment with expert ratings.

The reviewers and I see the main strength of the paper as shifting evaluation from simple classification accuracy to clinically relevant narrative dimensions like Safety and Reasoning Coherence. The reviewers also liked "clinician-in-the-loop" process. However, reviewers pointed out several weaknesses: (1) reliance on model-drafted references (2) single data source limiting demographic diversity, and (3) lack of clarity regarding the evaluation rubric and method workflow. (4) The use of "LLM-as-a-Judge" aspect also raised some questions. Overall I think there are still a lot of issues about presentation and methodology and experiments that need to be addressed clearly. IMO the paper can not be accepted in its current form. However, this is a great initiative. I encourage resubmission to a future ML venue after a comprehensive revision.

**Reviewer Concerns:**

The authors I think alleviated concerns about methodological clarity, standardization and redesigning complex figures. They also clarified the human evaluation protocol. The necessity of the RL is less clear. The anchor bias of revising rather than writing from scratch is a major limitation also. The lack of diversity of the data is also a valid concern.

**Reviewer Scores:**

Reviewer 69Ao (Current: 4): Likely to keep. The authors partially addressed the primary complaints about figure readability and the lack of reliability experiments, and improving visual presentation. They did not however address other concerns such as lack of sufficient experimentation.
Reviewer fDQ7 (Current: 4): Likely to keep or increase 5. The concern about "model-drafted references" is a major issue. The authors’ response may have alleviated concerns about certification process and the scalability trade-offs..
Reviewer jSyH (Current: 2): Likely to increase to 3. Improvement in human eval and key eval. Necessity of RL and training strategy is less clear would have convinced the reviewer.
Reviewer TBj2 (Current: 6): Keep at 6.

---

### Decision · Program_Chairs · 2026-01-26

Reject